# Effects of Early Life Stress on Epigenetic Changes of the Glucocorticoid Receptor 1_7_ Promoter during Adulthood

**DOI:** 10.3390/ijms21176331

**Published:** 2020-08-31

**Authors:** Mi Kyoung Seo, Seon-gu Kim, Dae-Hyun Seog, Won-Myong Bahk, Seong-Ho Kim, Sung Woo Park, Jung Goo Lee

**Affiliations:** 1Paik Institute for Clinical Research, Inje University, Busan 47392, Korea; banaba66@inje.ac.kr; 2Department of Psychiatry, College of Medicine, Haeundae Paik Hospital, Inje University, Busan 47392, Korea; seon0827@naver.com; 3Department of Biochemistry, Department of Convergence Biomedical Science, College of Medicine, Dementia and Neurodegenerative Disease Research Center, Inje University, Busan 47392, Korea; daehyun@inje.ac.kr; 4Department of Psychiatry, College of Medicine, The Catholic University of Korea, Seoul 07345, Korea; wmbahk@catholic.ac.kr; 5Department of Internal Medicine, College of Medicine, Haeundae Paik Hospital, Paik Institute for Clinical Research, Inje University, Busan 47392, Korea; junjan@paik.ac.kr; 6Department of Convergence Biomedical Science, College of Medicine, Paik Institute for Clinical Research, Department of Health Science and Technology, Graduate School, Inje University, Busan 47392, Korea; 7Department of Psychiatry, College of Medicine, Haeundae Paik Hospital, Paik Institute for Clinical Research, Department of Health Science and Technology, Graduate School, Inje University, Busan 47392, Korea

**Keywords:** depression, early life stress, epigenetic, glucocorticoid receptor, histone modification, hippocampus

## Abstract

Growing evidence suggests that early life stress (ELS) has long-lasting effects on glucocorticoid receptor (GR) expression and behavior via epigenetic changes of the GR exon 1_7_ promoter. However, it remains unclear whether ELS regulates histone modifications of the GR exon 1_7_ promoter across the life span. We investigated the effects of maternal separation (MS) on histone acetylation and methylation of GR exon 1_7_ promoter in the hippocampus, according to the age of adults. Depression-like behavior and epigenetic regulation of GR expression were examined at young and middle adulthood in mice subjected to MS from postnatal day 1 to 21. In the forced swimming test, young adult MS mice showed no effect on immobility time, but middle-aged MS mice significantly increased immobility time. Young adult and middle-aged MS mice showed decreased GR expression. Their two ages showed decreased histone acetylation with increased histone deacetylases (HDAC5) levels, decreased permissive methylation, and increased repressive methylation at the GR exon 1_7_ promoter. The extent of changes in gene expression and histone modification in middle adulthood was greater than in young adulthood. These results indicate that MS in early life causes long-term negative effects on behavior via histone modification of the GR gene across the life span.

## 1. Introduction

Early life stress (ELS) refers to various types of abuse by parents or caregivers that children and adolescents may experience such as child abuse, neglect, and loss [1]. Children who experienced ELS have a three-to-fourfold higher risk of depression than children who did not [2]. The severity of symptoms and the rate of recurrence are also high in these children [3,4].

Many human and animal studies have verified that ELS causes epigenetic changes in the glucocorticoid receptor (GR) gene (NR3C1), which regulates gene expression and behavior in adults [5,6,7]. The 5′ noncoding region of the GR gene in rodents consists of 11 untranslated first exons (I1–I11), each of which has its own promoter [8]. Among the first exon promoters, the exon 1_7_ (exon IF homolog in human) promoter is responsive to DNA methylation induced by ELS, which causes a decrease in GR expression in rodents and humans [9]. At this time, most studies on epigenetic changes in the GR exon 1_7_/1_F_ promoter induced by ELS have focused on DNA methylation.

Many studies on epigenetic changes in the brain and periphery in depressed humans or after chronic stress in animal models support the important role of histone modification (i.e., histone acetylation and methylation) [10]. Histone modification at specific gene promoters control gene activation or silencing by altering the chromatin structure and compaction [11,12]. Histone acetylation, which occurs at the lysine (K) residue of the histone tail, loosens the bond between DNA and histone protein to facilitate bonding between RNA polymerase and transcription activator, thus activating gene expression [10]. The balance of histone acetylation is regulated by histone acetyltransferases (HATs) and histone deacetylases (HDACs). HATs increase gene expression by promoting histone acetylation, while HDACs decrease gene expression by promoting histone deacetylation [11]. On the other hand, histone methylation plays a role in either activating (H3K4 and H3K36) or repressing (H3K9, H3K27, and H3K20) gene expression, depending on the K and the valence state (mono-, di-, or trimethylation) where it occurs [11].

Despite the contribution of histone modification to depression, studies on histone modification of the GR gene in early adverse experiences remain limited. Studies in rats have suggested that the quality of maternal care is associated with histone acetylation at the exon 1_7_ promoter of GR [13,14]. In this study, the offspring of mothers that showed high lick-groom and arched-back nursing (LG-ABN) behaviors showed increased GR expression, a more modest response to stress, and increased acetylation of histone H3 (H3K9) at the GR promoter 1_7_ compared with the offspring of low LG-ABN mothers [13,14]. On the contrary, adult mice with an experience of maternal separation (MS) showed anxiety, depression-like behaviors, and hypothalamus-pituitary-adrenal axis dysfunction in adulthood [15,16,17], and these mice exhibited reduced hippocampal GR expression [18]. Our previous study demonstrates that the reduction of GR expression induced by MS was associated with reduced histone H3 acetylation at the GR promoter 1_7_ [7]. In the present study, we sought to further investigate changes in histone methylation as well as histone acetylation of the GR promoter I_7_. Although many studies have been reported that ELS induces epigenetic changes in the GR exon 1_7_ promoter [7,9,19], it remains unclear whether ELS regulates histone modification of the GR gene across the life span. Thus, we investigated depression-like behavior, hippocampal total GR expression, and histone acetylation and methylation at the GR promoter 1_7_ in young and middle adulthood across the life span in mice subjected to MS.

## 2. Results

### 2.1. Behavioral Changes in the Forced Swimming Test

Figure 1 represents the animal experimental design applied in the present study. We investigated the effect of MS on depression-like behavior using forced swimming test (FST) in young adults and middle-aged MS mice and their age-matched controls (Figure 2). There was no significant difference in immobility times between young adult controls and young adult MS mice (75.58 ± 15.15 s vs. 63.74 ± 15.85 s; t = 0.533, *p* = 0.599). However, the depression-like phenotype was observed to have increased the immobility time significantly for middle-aged MS mice compared to middle-aged controls (26.38 ± 3.30 s vs. 59.81 ± 10.87 s; t = 3.199, *p* = 0.008).

Mouse pups were separated from their mothers from postnatal day (PND) 1 to 21 for 3 h daily. When the mouse pups reached young (PND 60) or middle (PND 240) adulthood, they were subjected to the forced swimming test (FST), respectively. After FST, the hippocampi from the whole brain were dissected for further analysis.

### 2.2. Changes in Total GR mRNA Expression

We next examined whether MS mice across the life span exhibit changes in hippocampal total GR expression. For this, reverse transcription and qRT-PCR were performed from young adult and middle-aged MS mice and their age-matched controls (Figure 3). There was a significant MS effect on the levels of the total GR mRNA in the hippocampus of both young (1.00 ± 0.09 vs. 0.80 ± 0.06; *t* = 3.304, *p* = 0.007) and middle adulthood (1.00 ± 0.14 vs. 0.19 ± 0.13; *t* = 12.300, *p* < 0.001) mice. Moreover, MS animals showed an age-dependent decrease in the total GR level in middle adulthood.

### 2.3. Changes of Histone H3 Acetylation of the GR Exon 1_7_ Promoter

Given that MS decreases hippocampal GR expression across the life span, we investigated whether MS has an age-dependent effect on histone H3 acetylation at GR promoter 1_7_. The levels of histone H3 acetylation were analyzed using the ChIP assay with the global histone H3 acetylation (K9 + K14) antibody. Animals with early stress of MS showed a significant reduction in acetylated histone H3 level of GR promoter 1_7_ at young (1.00 ± 0.05 vs. 0.76 ± 0.13; *t* = 2.206, *p* = 0.017) and middle adulthood (1.00 ± 0.11 vs. 0.28 ± 0.09; *t* = 13.460, *p* < 0.001, Figure 4A). Similar to those results observed for GR levels, the MS effect on histone acetylation was more apparent in middle adulthood than in young adulthood.

We further investigated whether the expression of HDAC5, one of the class II HDACs, was regulated by MS. Enhanced HDAC5 expression was observed in the hippocampus of young adult and middle-aged MS animals compared to their age-matched controls (young adulthood: 1.00 ± 0.13 vs. 1.43 ± 0.12, *t* = 2.979, *p* = 0.007; middle adulthood: 1.00 ± 0.06 vs. 1.80 ± 0.07, *t* = 8.466, *p* = 0.003, Figure 4B). The extent of increase was greater in middle adulthood than in young adulthood.

### 2.4. Changes of Histone H3 Methylation of the GR Exon 1_7_ Promoter

We then investigated whether MS regulates permissive histone methylation (trimethylation of H3K4, H3K4me^3^), and repressive histone methylation (trimethylation of H3K27, H3K27me^3^) of the GR promoter 1_7_ in the hippocampus of young and middle adulthood. A strong decrease in permissive H3K4me^3^ in MS mice was observed in both young (1.00 ± 0.16 vs. 0.51 ± 0.15; *t* = 4.470, *p* = 0.001) and middle adulthood (1.00 ± 0.05 vs. 0.39 ± 0.21; *t* = 5.907, *p* < 0.001, Figure 5A). On the other hand, MS mice showed an enhanced repressive H3K27me^3^ in both young (1.00 ± 0.14 vs. 1.45 ± 0.18; *t* = 2.279, *p* = 0.034) and middle adulthood (1.00 ± 0.10 vs. 1.82 ± 0.16; *t* = 4.370, *p* < 0.001, Figure 5B). Changes in histone methylation induced by MS were higher in middle adulthood than in young adulthood.

## 3. Discussion

In this study, we investigated how postnatal MS affects GR expression and epigenetic modifications in young and middle adulthood in mice. We also observed that MS reduced mRNA expression based on changes in histone H3 acetylation and methylation of the GR exon 1_7_ promoter during the young and middle adulthood of mice. However, we did not find a significant difference in the immobility time between the MS and control group in young adulthood, while the immobility time increased in the MS group of middle adulthood. Overall, in the MS group, the phenotype of depression appears with age and growth.

In animal model studies using MS as ELS that have been reported so far, it remains unclear how MS affects behavioral phenotypes. Millstein et al. reported that repeated MS did not produce an anxiety-related phenotype or depression-related phenotype in mice [20]. Rüedi-Bettschen et al. reported a decrease in immobility time in the MS group [21]. Lambás-Señas et al. reported that MS was administered for 3 h every day from PND 2 to 15 in Sprague–Dawley rats, and that immobility time increased when rats became adults [22]. In our study, we observed that rats were subjected to MS every 3 h per day from PDN 1 to 21, and the immobility time increased when rats became adults [23]. We also showed that brain-derived neurotrophic factor (BDNF) and GR are involved in this mechanism, similar to previous studies [7,23,24].

It is known that phenotypes of depression could be observed in GR knockout mice; GR knockout mice have a smaller sucrose intake compared to normal mice, and blood glucocorticoid levels are higher than in normal mice. If dexamethasone is administered to normal mice, glucocorticoid production is inhibited. However, when dexamethasone is administered to a GR knockout mouse, there is no change in the glucocorticoid level. Therefore, changes in the activity of the GR gene due to the GR gene mutation can cause depression [25]. In this study, we confirmed that the occurrence of depressive phenotype due to MS was not observed in young adulthood, but GR expression was reduced. However, in middle adulthood, a depressive phenotype was observed, and it was confirmed that GR expression was also reduced. Navailles et al. observed that mice subjected to MS (PND 2–15; 3 h per day) showed reduced expression of GR mRNA in the forebrain during adolescence and adulthood. In addition, they confirmed that reduced GR mRNA expression was restored when the antidepressant drug fluoxetine was administered [26]. In addition, Ladd et al. reported a decrease in the expression of GR mRNA in the frontal cortex and hippocampus of adult mice experiencing MS (PND 2–14: 3 h per day), which is consistent with the results of this study [18]. However, even though a decrease in GR mRNA expression was due to a change in histone modification during adolescence, GR expression in adulthood might not affect the depressive phenotype. This is because MS can affect other genes in various ways, causing neuroendocrine changes and molecular changes. Therefore, it is important to analyze the pattern of depressive behavior more accurately, and the sucrose preference test could be used to analyze depressive behavior better than the FST used in this study. Therefore, in the future, studies using methods such as the sucrose preference test will be required.

To determine whether histone modification is involved in GR mRNA reduction due to ELS, we examined histone H3 acetylation and the methylation of the GR exon 1_7_ promoter for different life-stages. Transcription-activating histone H3 acetylation and histone H3K4me^3^ show a significant decrease and transcription-inhibiting histone H3K27me^3^ has increased. This indicates that environmental stress factors are related to GR mRNA reduction by negatively influencing histone modification. There have been many studies on the epigenetic mechanisms regarding the GR exon 1_7_ promoter focused on DNA methylation rather than histone modification [9,27]. The GR exon 1_7_ promoter with the NGF1A binding site, DNA methylation of this region, and the relationship between depression and ELS were the focus of most studies [28].

Studies using rodents to elucidate the epigenetic mechanism of GR genes have revealed that the difference between DNA methylation and histone H3 acetylation of the GR exon 1_7_ promoter is caused by quality differences in the postnatal maternal care of mouse pups [13,14]. After giving birth, mouse mothers generally show behaviors such as licking their pups, grooming their pups’ fur, and arching the back to nurse their pups (LG-ABN behaviors). Methylation of the GR exon 1_7_ promoter was reduced and histone H3 acetylation was increased in mouse pups to which the mother showed a high level of LG-ABN behaviors compared to those whose mother did not [13,14]. Based on these findings, McGowan et al. conducted the first human subject study regarding the GR exon 1_F_ promoter, which is similar in structure to the GR exon 1_7_ promoter in rodents. They observed that cytosine methylation of the NGF1A binding site of the GR exon 1_F_ promoter increased, and the levels of GR mRNA decreased in postmortem hippocampus of suicide victims with a history of child abuse [29]. In another study, after chronic social defeat, stress was applied to mice, antidepressant imipramine was administered, and histone modification of the BDNF promoter in the hippocampus was observed. Transcription-inhibiting histone H3K27me^2^ was reduced by imipramine while transcription-increasing histone H3K4me^2^ and acetylation of histone H3 at K9 and K14 were increased, and BDNF expression was increased due to histone modification [30].

Several studies have shown that HDAC affects the pathophysiology of depression [10,30,31]. In particular, HDAC5 may be associated with depression in several types of HDAC subtypes. Tsankova et al. observed that HDAC5 expression was reduced in the hippocampus of socially defeated mice treated with the antidepressant imipramine [30]. In addition, Rentathal et al. reported that HDAC5 expression was reduced, and depressive behavior appeared in the nucleus accumbens of socially defeated mice, and it was also observed that chronic social defeat stress increased depression-like behaviors in HDAC5 knockout mice [32]. In a clinical study, HDAC5 expression was found to have increased in the blood of patients with depression [33]. In this study, it was observed that ELS increased HDAC5 mRNA expression in young and middle adulthood. These findings indicate that increases in HDAC5 expression should be associated with both the preclinical and clinical development of depression.

There are several limitations to this study that should be noted. First, we did not examine changes in levels of GR exon 1_7_ mRNA unique to the GR exon 1_7_ promoter. According to previous studies [7,14,34], reduction of hippocampal exon 1_7_ GR mRNA levels induced by ELS was associated with reduced levels of GR mRNA or protein. For this reason, we only investigated GR mRNA levels. However, for a detailed study on transcriptional regulation from the GR promoter 1_7_, changes in exon 1_7_ mRNA levels should be investigated. In addition, it is important to examine at least one other exon of GR or another gene that are unchanged. Second, we performed FST to characterize the depression-like phenotype. Although many researchers use this model for behavioral despair tests, the FST is a more appropriate tool for screening antidepressant drugs [35]. Therefore, additional behavioral studies require a sucrose preference test, which is a measure of anhedonia, or the Morris water maze test, which is a measure of spatial memory. Third, we did not examine HDAC5 protein levels. Additional experiments for protein levels are required to support changes in histone acetylation levels.

In conclusion, MS, which is experienced early in life, can cause a depressive phenotype by inducing continuous genetic modifications of the GR gene through epigenetic mechanisms. In addition, harmful experiences at the early-life stage can affect the development of psychiatric diseases, such as depression in adulthood.

## 4. Materials and Methods

### 4.1. Animals

All experimental procedures were conducted in accordance with guidelines for the care and use of laboratory animals for scientific purposes with approved protocols from the Committee for Animal Experimentation and the Institutional Animal Laboratory Review Board of Inje Medical College (approval no. 2016-053). Pregnant female C57BL/6J mice were purchased from Daehan Biolink (Chungbuk, Korea) on gestation day 15. Pregnant females were individually housed with sawdust until delivery. All animals were housed in a temperature (21 ± 2 °C) and humidity (50 ± 5%) controlled room, and maintained on a 12-h light/dark cycle with food and water provided *ad libitum*.

### 4.2. Maternal Separation (MS)

Litters from dams were divided randomly to control and MS groups. Pups from MS groups were separated from the dam for 3 h daily from PND 1 to PND 21. After separation, the pups were returned to their home cage and to their dam. Pups from control groups remained undisturbed with their dam. Male mice were examined at two adult ages (PND 60 in young adulthood, and PND 240 in middle adulthood)—Young adult control mice (CON group) and young adult MS mice (MS group) in young adulthood; middle-aged control mice (CON group) and middle-aged MS mice (MS group) in middle adulthood (Figure 1).

### 4.3. Forced Swimming Test (FST)

The FST method was performed as described by Porsolt et al. [36], with minor modifications. Briefly, the control and MS groups were tested in young and middle adulthood, respectively. The mouse was gently placed in custom-made plastic cylinders (25 × 10 cm) containing 12 cm of water (23 ± 2 °C), and the immobility times were recorded for 7 min. The first 2 min was a habituation period for swimming, and the total duration of immobility was measured in the remaining 5 min.

### 4.4. Measurement of mRNA Levels Using Quantitative Real-Time (qRT-PCR)

After FST, the whole mouse brain from all mice was removed after decapitation. The hippocampus was dissected from the brain on ice, rapidly frozen in liquid nitrogen, and stored at −80 °C until required for further experiments. Total RNA of the hippocampus was extracted with Qiazol reagent (Qiagen, Valencia, CA, USA). Total RNA was frozen at −80 °C until use. Complementary DNA (cDNA) synthesis was performed with 1 μg of each RNA sample and reverse transcribed to cDNA using the amfiRivert II cDNA Synthesis Master Mix (GenDepot, Barker, TX, USA). qRT-PCR was performed using 100 ng of cDNA, 10 pmole of each primer, and TOPreal™ qPCR 2X PreMIX (SYBR Green with low ROX; RT500, Enzynomics, Daejeon, Korea), in an ABI 7500 (Applied Biosystems, Foster City, CA, USA). Amplification conditions were 95 °C for 10 min followed by 40 cycles of 95 °C for 15 s, 55 °C for 35 s, and 72 °C for 35 s. The primer sequences for total GR, HDAC5, and glyceraldehyde-3-phosphate dehydrogenase (GAPDH) are provided in Table 1. All data were analyzed using the 2^−△△Ct^ method, and normalized to GAPDH.

### 4.5. Chromatin Immunoprecipitation Assay (ChIP Assay)

The ChIP assay was used from the SimpleChIP^®^ Plus Enzymatic Chromatic IP kit (#9005, Cell Signaling, Beverly, MA, USA) according to the manufacturer’s instructions and as described previously [7,23,24]. Briefly, the DNA-protein of hippocampal tissue was crosslinked with 1.5% formaldehyde for 20 min at room temperature. The crosslinking reaction was stopped by adding glycine. Chromatin was then dissolved in lysis buffer containing a protease inhibitor cocktail (PIC; Roche, Indianapolis, IN, USA) and sonicated using EpiShear™ Probe Sonicator (Active Motif, Carlsbad, CA, USA) for 10 min on ice to produce fragments of 200–500 bp. After removal of an aliquot as an input DNA, the samples were incubated overnight at 4 °C in assay tubes containing the following antibodies: Anti-histone H3 acetylated at K9 and K14 (AcH3; 06-599, Millipore, Billerica, MA, USA), anti-histone H3 trimethylated on K4 (H3K4me^3^; ab8580, abcam, Cambridge, MA, USA), or anti-histone H3 trimethylated on K27 (H3K27me^3^; ab6002, abcam, Cambridge, MA, USA). Three antibodies used in the present study were based on previous studies [30,37]. The immunoprecipitations were pulled down using the ChIP-grade protein G magnetic beads and washed with a low and high salt buffer. The DNA-histone protein crosslinks were eluted with elution buffer and de-cross-linked with proteinase K digestion at 65 °C. The resulting DNA was processed using phenol-chloroform (Amresco, Solon, OH, USA) extraction and ethanol precipitation. The purified DNA was then subjected to qRT-PCR. qRT-PCR was performed using DNA, 10 pmole of primer, and TOPreal™ qPCR 2X PreMIX (SYBR Green with low ROX; RT500, Enzynomics, Daejeon, Korea) in an ABI 7500. Amplification conditions were 95 °C for 10 min followed by 45 cycles of 95 °C for 15 s, 55 °C for 35 s, and 72 °C for 35 s. The primer sequences for GR exon 1_7_ promoter are provided in Table 1. All data were analyzed using the 2^−△△Ct^ method and normalized to input DNA.

The HDAC5 and GR exon 1_7_ primers used in this study were designed with the Primer3 software (version 0.4.0, Whitehead Institute, MT center for Genome research). GAPDH: Glyceraldehyde 3-phosphate dehydrogenase; GR: Glucocorticoid receptor; HDAC5: histone deacetylases 5.

### 4.6. Statistical Analysis

Comparison of the control and MS groups in young or middle adulthood were analyzed using the *unpaired student t-test* (Prism 8.0, Graphpad Software Inc., La Jolla, CA, USA). The data shown are the mean ± standard error of the mean (SEM), and the difference was considered to be statistically significant when *p* < 0.05.

## Figures and Tables

**Figure 1 ijms-21-06331-f001:**
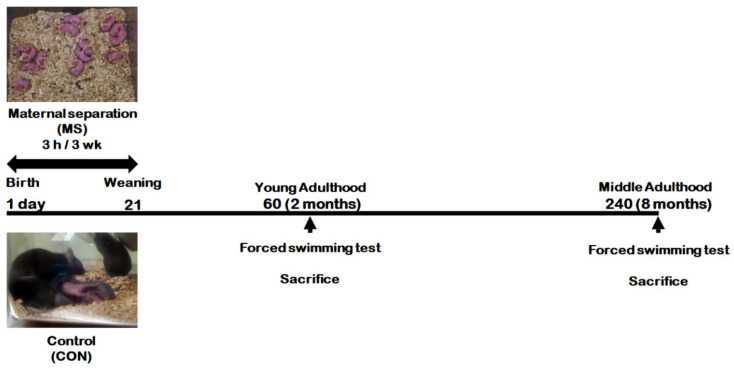
Schematic representation of the experimental design.

**Figure 2 ijms-21-06331-f002:**
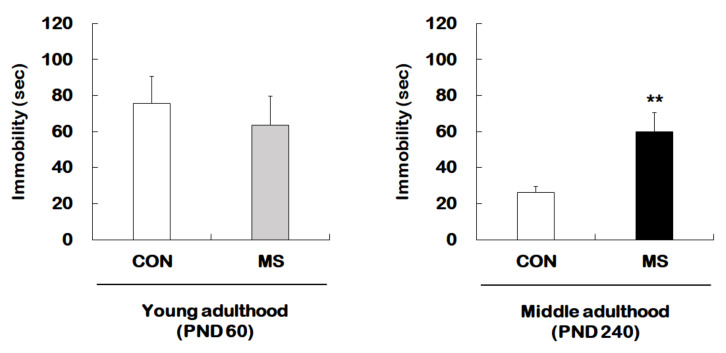
Effects of maternal separation (MS) on depression-like behavior in young and middle adulthood. Animals with MS and their age-matched controls were tested in young (PND 60) and middle (PND 240) adulthood using the FST. Data are expressed as the mean ± SEM (*n* = 7–13/group, ** *p* < 0.01, unpaired Student *t*-test).

**Figure 3 ijms-21-06331-f003:**
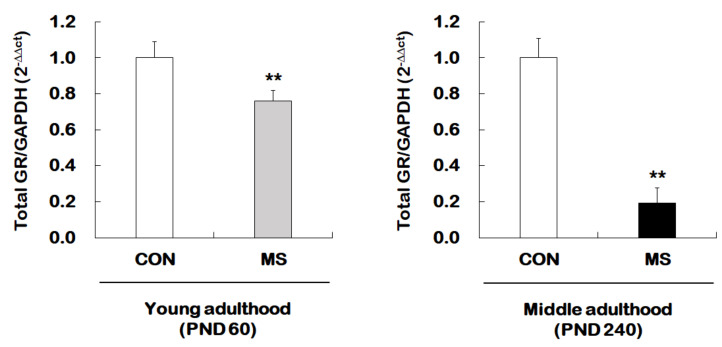
Age-dependent changes in total glucocorticoid receptor (GR) expression following MS exposure. Levels of total GR mRNA were analyzed in young adulthood (PND 60) and middle (PND 240) adulthood mice exposed to MS using qRT-PCR. All quantities were normalized to GAPDH. Data are expressed as values relative to the CON group using the 2^−△△ct^ method, and represent the mean ± SEM (*n* = 10–13/group, ** *p* < 0.01, unpaired Student *t*-test).

**Figure 4 ijms-21-06331-f004:**
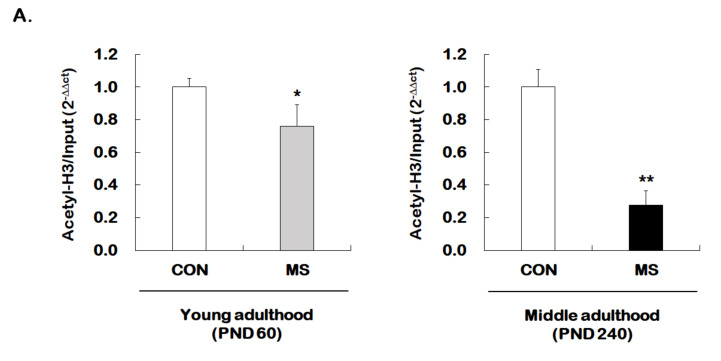
Age-dependent changes in the level of histone H3 acetylation at the GR promoter 1_7_ and HDAC5 expression following MS exposure. (**A**) ChIP assays were used to measure the level of acetylated histone H3 (Acetyl-H3) at the GR promoter 1_7_ in the hippocampus using an antibody to Acetyl-H3. Data were normalized to input DNA (*n* = 10–13/group). (**B**) The HDAC5 mRNA level in the hippocampus was measured using qRT-PCR. All quantities were normalized to GAPDH (*n* = 9–13/group). These levels were analyzed in young (PND 60) and middle (PND 240) adulthood, respectively. Data were expressed as values relative to the CON group using the 2^−△△ct^ method and represent the mean ± SEM (* *p* < 0.05, ** *p* < 0.01, unpaired Student t-test).

**Figure 5 ijms-21-06331-f005:**
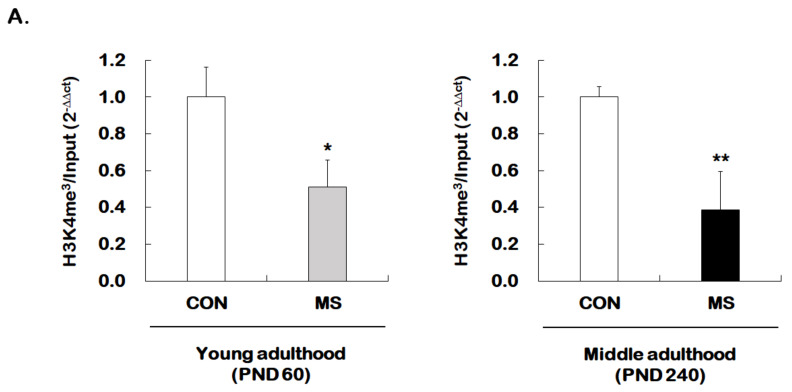
Age-dependent changes in the levels of histone H3 methylation at the GR promoter 1_7_ following MS exposure. ChIP assays were used to measure the level of acetylated histone H3 trimethylated K4 (H3K4me^3^, **A**) and K27 (H3K27me^3^, **B**) at GR promoter 1_7_ in the hippocampus using antibodies to H3K4me^3^ and H3K27me^3^. These levels were analyzed in young (PND 60) and middle (PND 240) adulthood, respectively. Data were normalized to input DNA and are expressed as values relative to the CON group using the 2^−△△ct^ method, and represent mean ± SEM (*n* = 9–13/group, * *p* < 0.05, ** *p* < 0.01, unpaired Student *t*-test).

**Table 1 ijms-21-06331-t001:** Primers used in this study.

Gene Name	Gene Bank Number	Primer Sequence (5′-3′)	
quantitative real-time PCR (qRT-PCR)			
Total GR [38]	NM_008173.3	Forward	AGGCCGCTCAGTGTTTTCTA
		Reverse	TACAGCTTCCACACGTCAGC
HDAC5	NM_001284249.1	Forward	CCATTGGAGATGTGGAATAC
		Reverse	CAGTGGAGACAGATGTCCTT
GAPDH [39]	NG_007244.3	Forward	AACAGCAACTCCCATTCTTC
		Reverse	TGGTCCAGGGTTTCTTACTC
qRT-PCR for histone modification			
GR exon 1_7_ promoter	XM_006525663.4	Forward	TTCCGTGCCATCCTGTA
		Reverse	CCGAGTTTCTTTAGTTTCTCTT

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
