# Peer review of "Effects of Early Life Stress on Epigenetic Changes of the Glucocorticoid Receptor 17 Promoter during Adulthood"

_ijms, 2020, doi:10.3390/ijms21176331_

Round 1
Reviewer 1 Report
Seo et al. provide an interesting examination of how early life maternal separation can induce lifelong changes in histone acetylation and methylation of the glucocorticoid receptor gene (GR). They specifically show that early life maternal separation represses GR expression at mid and adult timepoints and that these expression changes are associated with altered histone modifications. The authors further suggest HDAC5 as one potential mediator of the epigenetic changes induced by maternal separation. While interesting, there are several major and minor concerns that need to be addressed prior to publication.
Major Concerns:
- For both the histone acetylation and histone methylation ChIP-qPCR experiments, are the changes in GR levels unique to exon I7 of the GR? It is critical to show at least one other site in the GR or another gene that is unchanged. It would also be interesting to look at the BNDF gene that has also been linked to similar stress-related regulation.
- There are several areas in the methods that are unclear.
- For example, the methods description for the ChIP is not sufficient. Please provide actual experimental details, not simply a paper citation.
- The description of the groups on lines 294-301 is confusing. They read as if MS was applied on PND 60 or PND 240; however, I think the authors did the MS always on PND1-21 and the animals were tested on PND60 or 240. This is not clear from the text. The figure in the methods would also be better served as panel A of Figure 1.
- It is unclear what the authors mean on line 292 in reference to the animal breeding: “we bred only male mice.” I assume that means they tested only male mice? What is the rationale to justify this?
- It is a bit unclear when t-tests versus ANOVAs are used. Please report the specific test, t or F values and the actual p values for each relevant comparison. If they are t-tests then the graphs for the two ages (PND60 and PND240) should be separate panels. If it is an ANOVA with corrected posthoc t-test then they can stay as one panel but the ANOVA and t-test results should be reported.
- Are the same animals used for behavior assessment as for the molecular experiments? How do we know that the behavior didn’t influence the epigenetic differences?
- What is the rationale for looking at HDAC5? It isn’t clear from the text why you are looking at HDAC5 mRNA and HDAC5 binding (ChIP) until the reader gets to the end of the discussion. What about other HDACs or HATs? Are the HDAC5 mRNA levels also observed at the protein level? It would be nice to have both the HDAC5 ChIP qPCR and western quantification of HDAC5 protein levels.
Minor Concerns:
- There are several textual inaccuracies that need correction. For example, on lines 84-86, the histones do not “bind” methyl groups. Instead the sentence should read that the lysines on the histone tails are modified by the chemical addition of methyl groups. Similarly, the use of the term “bad parenting” (line 60) is subjective and inappropriate as a description of early life stress. Consider using “abuse” or another more scientific term. On line 107, the paragraph begins with “Studies about DNA epigenetic mutation”. I think the authors mean epigenetic modifications, as mutation does not make sense given the context. Also, on line 117, the authors state they are examining the “development of depression”. Depression is a human disorder, they are modeling aspects of depression and should be clear about this distinction throughout the text (ie. lines 372-373). Similarly, on line 185-187, the methyl groups are listed as “bonded” to the histones. Please use chemical modification.
- Please check the wording on lines 376-378. This sentence does not make sense. There are also several grammatical mistakes throughout the text, particularly in the introduction and discussion that make the text very difficult to follow. Please have a native English speaker edit the final version before publication.
- On line 253, mice do not have hair, they have fur.
Author Response
Summary of changes made
We appreciate the Reviewers feedback on our manuscript. All suggested corrections have been made.
Reviewer: 1
- For both the histone acetylation and histone methylation ChIP-qPCR experiments, are the changes in GR levels unique to exon I7 of the GR? It is critical to show at least one other site in the GR or another gene that is unchanged. It would also be interesting to look at the BNDF gene that has also been linked to similar stress-related regulation.
> This comment brings up an important issue. This issue was described as ‘limitations of this study’ in the Discussion section (line 494 to 500 of page 15). As suggested by the Reviewer, it would also be interesting to investigate stress-related genes, such as the BDNF gene. We are assessing another gene, p11 (S100A10), which will be presented in a future report.
- There are several areas in the methods that are unclear.
> 4. The Materials and Methods section has been revised in the modified manuscript.
(1) For example, the methods description for the ChIP is not sufficient. Please provide actual experimental details, not simply a paper citation.
> As suggested, the description for the ChIP assay was rewritten in the revised manuscript.
(2) The description of the groups on lines 294-301 is confusing. They read as if MS was applied on PND 60 or PND 240; however, I think the authors did the MS always on PND1-21 and the animals were tested on PND60 or 240. This is not clear from the text. The figure in the methods would also be better served as panel A of Figure 1.
> The description of the groups was rewritten and Figure 5 was changed to Figure 1 in the revised manuscript.
(3) It is unclear what the authors mean on line 292 in reference to the animal breeding: “we bred only male mice.” I assume that means they tested only male mice? What is the rationale to justify this?
> Males are commonly used in animal experiments for depression. Data variability in females is large due to disturbing factors, such as the menstrual cycle (Ref).
Ref; Barbara Planchez, Alexandre Surget & Catherine Belzung. Animal models of major depression: drawbacks and challenges. Psychiatry and Preclinical Psychiatric Studies. 04 October 2019
(4) It is a bit unclear when t-tests versus ANOVAs are used. Please report the specific test, t or F values and the actual p values for each relevant comparison. If they are t-tests then the graphs for the two ages (PND60 and PND240) should be separate panels. If it is an ANOVA with corrected posthoc t-test then they can stay as one panel but the ANOVA and t-test results should be reported.
> As suggested, t values were added in the Results section. The graphs for young adulthood and middle adulthood were separated, respectively.
- Are the same animals used for behavior assessment as for the molecular experiments? How do we know that the behavior didn’t influence the epigenetic differences?
> The same animals used for the FST were assessed for molecular experiments. The mice were sacrificed immediately after FST. Thus, we assume that one behavioral test did not affect the epigenetic differences over a short time.
- What is the rationale for looking at HDAC5? It isn’t clear from the text why you are looking at HDAC5 mRNA and HDAC5 binding (ChIP) until the reader gets to the end of the discussion. What about other HDACs or HATs? Are the HDAC5 mRNA levels also observed at the protein level? It would be nice to have both the HDAC5 ChIP qPCR and western quantification of HDAC5 protein levels.
> The rationale for focusing on HDAC5 in this study was addressed in the Discussion section (line 483 to 491 of page 14 to 15). Our research group aimed to investigate HATs activating transcription in future studies. A limitation of this study was that the levels of HDAC5 protein were not examined; this issue was addressed in the Discussion section (line 504 to 505 of page 15) and we plan to investigate this issue in a future study.
- There are several textual inaccuracies that need correction. For example, on lines 84-86, the histones do not “bind” methyl groups. Instead the sentence should read that the lysines on the histone tails are modified by the chemical addition of methyl groups. Similarly, the use of the term “bad parenting” (line 60) is subjective and inappropriate as a description of early life stress. Consider using “abuse” or another more scientific term. On line 107, the paragraph begins with “Studies about DNA epigenetic mutation”. I think the authors mean epigenetic modifications, as mutation does not make sense given the context. Also, on line 117, the authors state they are examining the “development of depression”. Depression is a human disorder, they are modeling aspects of depression and should be clear about this distinction throughout the text (ie. lines 372-373). Similarly, on line 185-187, the methyl groups are listed as “bonded” to the histones. Please use chemical modification.
> These issues were deleted or corrected by revising the entire manuscript.
- Please check the wording on lines 376-378. This sentence does not make sense. There are also several grammatical mistakes throughout the text, particularly in the introduction and discussion that make the text very difficult to follow. Please have a native English speaker edit the final version before publication.
> The English in this document has been checked by at least two professional editors, both native speakers of English. For a certificate, please see: http://www.textcheck.com/certificate/ Jgq5Po
- On line 253, mice do not have hair, they have fur.
> This issue was deleted or corrected by revising the entire manuscript.
Reviewer 2 Report
The manuscript by Seo et al. presents new data regarding the effects of daily maternal separation from birth through weaning on the expression of Nr3c1 mRNA levels in the hippocampus of C57BL6 mice. In addition, these mice were also examined for depression-like symptoms using the forced swim test at either young adulthood (~2 months of age) vs. middle adulthood (8 months of age). In addition, the authors examined H3K9/14 acetylation and H3K4 trimethylation levels proximal to the Nr3c1 17 promoter in the treated and control mice. While there is some new information presented in this paper, the Nr3c1 17 promoter is perhaps one of the most studied promoters ever. This is, in part, due to pioneering work by Michael Meaney and collaborators who first discovered a correlation between DNA methylation and the licking grooming behavior of mothers some years ago. The current paper examines the effects of maternal separation on glucocorticoid receptor expression and two histone modifications associated with the corresponding promoter.
While the paper reports some results of potential value, there are several factors which detract from the enthusiasm for the manuscript.
- There are numerous spelling errors, grammatical errors and inappropriate statements that need the attention of a good editor familiar with English. Second, the rationale for the current study is not adequately described.
- The glucocorticoid receptor has some 11 distinct promoters and the authors have only described results at one of these. Some rationale as to why the others were omitted should be provided. One might expect that epigenetic changes at other promoters would be different and this would provide some validation for changes observed at the 17 promoter.
- What was the rationale for choosing these particular histone modifications (i.e. H3K9/14 and H3K4me3)? Why wasn't DNA methylation examined in this paper. Since the amount of the GR mRNA goes down, one might anticipate that DNA methylation might increase as a result of the maternal separation.
- Another problem is that there are insufficient details provided to understand certain aspects of the study. For example, what antibodies were used for the ChIP experiments. Were these antibodies tested by the investigators for specificity.
- Why was HDAC5 chosen for this study. HDACs 1, 2, 3, and others also play major roles in the balance of acetylation/deacetylation levels. To select one of these and not the others, does not provide very compelling information regarding what's causing the observed changes. Moreover, the more appropriate measurement would be to examine Hdac5 protein levels and not mRNA. What about the changes in H3K4me3? Are there changes in the levels of these corresponding enzymes?
- In the presentation of the model in Section 4, the explanation of the of the Figure (Fig. 5) does not match the remainder of the text. For example, line 296: MS-young adulthood group: an experimental group in which animals experienced post-natal MS on PND 60. My impression from the body of the text is that the maternal separation always happened at the same time regardless of group. Instead, it was the time of the forced swim that varied. This description is very confusing.
- Finally, since mice are not natural born swimmers, did the authors consider an alternative behavioral test to obtain the relevant spatial memory information.
-
The statistical analysis section indicates that they used paired t-tests for the analysis. However, to make comparisons between groups, i.e. immobility times depending on whether the FST was performed in early vs, middle adulthood, would require use of 2-way ANOVA.
Author Response
Summary of changes made
We appreciate the Reviewers feedback on our manuscript. All suggested corrections have been made.
Reviewer: 2
- There are numerous spelling errors, grammatical errors and inappropriate statements that need the attention of a good editor familiar with English. Second, the rationale for the current study is not adequately described.
> The manuscript was completely revised. The English in this document has been checked by at least two professional editors, both native speakers of English. For a certificate, please see: http://www.textcheck.com/certificate/ Jgq5Po
- The glucocorticoid receptor has some 11 distinct promoters and the authors have only described results at one of these. Some rationale as to why the others were omitted should be provided. One might expect that epigenetic changes at other promoters would be different and this would provide some validation for changes observed at the 17 promoter.
> This issue was addressed in the Introduction section (line 137 to 143 of page 3)
- What was the rationale for choosing these particular histone modifications (i.e. H3K9/14 and H3K4me3)? Why wasn't DNA methylation examined in this paper. Since the amount of the GR mRNA goes down, one might anticipate that DNA methylation might increase as a result of the maternal separation.
> At this time, most studies on early life stress have focused on DNA methylation at the GR 17 promoter. However, histone modification is known to play an important role in the development of depression. Thus, we investigated changes in histone modification using representative markers on histone acetylation and methylation. These issues were addressed via revision of the Introduction section.
- Another problem is that there are insufficient details provided to understand certain aspects of the study. For example, what antibodies were used for the ChIP experiments. Were these antibodies tested by the investigators for specificity.
> 4. The Materials and Methods section was rewritten in detail. Information and references on antibodies for the ChIP assay were described.
- Why was HDAC5 chosen for this study. HDACs 1, 2, 3, and others also play major roles in the balance of acetylation/deacetylation levels. To select one of these and not the others, does not provide very compelling information regarding what's causing the observed changes. Moreover, the more appropriate measurement would be to examine Hdac5 protein levels and not mRNA. What about the changes in H3K4me3? Are there changes in the levels of these corresponding enzymes?
> The rationale for focusing on HDAC5 in this study was addressed in the Discussion section (line 483 to 491 of page 14 to 15). A limitation of the study was that the levels of HDAC5 protein were not examined; this issue was addressed in the Discussion section (line 504 to 505 of page 15) and we plan to investigate this issue in a future study. Studies on site- and sate-specific lysine methyltransferases are limited; enzymes corresponding to H3K4me3 and H3K27me3 were not investigated in the present study, but we plan to investigate this issue in a future report.
- In the presentation of the model in Section 4, the explanation of the of the Figure (Fig. 5) does not match the remainder of the text. For example, line 296: MS-young adulthood group: an experimental group in which animals experienced post-natal MS on PND 60. My impression from the body of the text is that the maternal separation always happened at the same time regardless of group. Instead, it was the time of the forced swim that varied. This description is very confusing.
> 4. The Materials and Methods section was rewritten in detail, including the legend of Figure 5 (Figure 5 was changed to Figure 1 in the revised manuscript).
- Finally, since mice are not natural born swimmers, did the authors consider an alternative behavioral test to obtain the relevant spatial memory information.
> This issue was described as a ‘limitation of this study’ in the Discussion section (line 500 to 504 of page 15).
- The statistical analysis section indicates that they used paired t-tests for the analysis. However, to make comparisons between groups, i.e. immobility times depending on whether the FST was performed in early vs, middle adulthood, would require use of 2-way ANOVA.
> Mouse pups were separated from their mothers from PND 1 to 21. When the mouse pups reached young adulthood (PND 60) or middle adulthood (PND 240), they were subjected to the forced swimming test (FST), respectively. After the FST, molecular experiments were performed from control vs. MS animals of young adulthood or middle adulthood. The statistical analysis used unpaired Student t test for comparison of control vs. MS according to the age of the adults. The graphs for young adulthood and middle adulthood were separated, respectively.
Round 2
Reviewer 2 Report
The revised manuscript by Seo et al., has improved in resubmission.
Point 1. The use of English, spelling and grammar have improved considerably. Because of these improvements, some of the questions previously raised have been addressed. However, other concerns still remain. In the Summary of Changes Made, the authors indicate their response to the previous review.
Point 2. With respect to why the authors only looked at promoter Nr3c1-7, the authors indicate that the issue was addressed on lines 137-143 of page 3. However, p. 3 of the manuscript has only 118 lines on it. However, I presume the reason was that they have already published another paper on histone modifications of the Nr3c1-7 promoter (Neurosci Lett, 2017). This is curious as in the 2017 paper, they also look at the effects of stress on the glucocorticoid receptor 1-7 promoter in terms of H3 histone acetylation. I point out, that other than this, these papers do not show much overlap. I have a general sense that the authors have responded to my critique without answering the specific questions directly. In some places, they indicate that these issues were addressed in the introduction. Needless to say, I was not able to find the requested information.
Point 3. I asked for the rationale for looking at H3K9/14ac and H3K4me3. They indicate that most people have looked at DNA methylation while histone modifications are also important. I agree, but was mainly wondering why those particular histone modifications were chosen.
Point 4. In response to my concern regarding their ChIP antibodies, the authors have rewritten the methods section and included additional detail. They include the manufacturer of the antibodies used. My question was – did they do any testing of these antibodies to detgermine their specificity or did they assume that the company sent them verified antibodies. They also mention that the magnetic beads are ChIP-grade. However, it’s generally of concern that the antibodies are ChIP grade, not the mag beads. The authors need to specific which part of the glucocorticoid receptor mRNA they are measuring. Is it the total NR3C1 mRNA or only the NR3C1-7 exon mRNA?
Point 5. I had previously requested why they chose to examine HDAC5. The authors did not respond appropriately. Did they only measure mRNA for HDAC5 or others as well. I am not sure that this measurement has any meaning in the absence of protein measurements for HDAC5. If, for example, if their housekeeping gene used for normalization changes in response to stress, the HDAC5 mRNA measurements will be off. If they want to make a connection between the decreased H3ac and HDAC5, it's the protein that would be responsible, not the mRNA.
Point 6. Movement of the Figure to Fig. 1 helps with my concerns. This is now clear.
Points 7 and 8 are adequately addressed.
Minor Concerns:
- In the legend to Fig. 3, changes is mis-spelled.
- In the legend to Fig. 4, acetylatuion is incorrectly spelled and there is a disconnect- ‘at the of GR.’
- In Table 1: it should be PCR for histone modification, not RT-PCR.
Author Response
We appreciate the Reviewers feedback on our manuscript. All suggested corrections have been made.
Point 1. The use of English, spelling and grammar have improved considerably. Because of these improvements, some of the questions previously raised have been addressed. However, other concerns still remain. In the Summary of Changes Made, the authors indicate their response to the previous review.
Point 2. With respect to why the authors only looked at promoter Nr3c1-7, the authors indicate that the issue was addressed on lines 137-143 of page 3. However, p. 3 of the manuscript has only 118 lines on it. However, I presume the reason was that they have already published another paper on histone modifications of the Nr3c1-7 promoter (Neurosci Lett, 2017). This is curious as in the 2017 paper, they also look at the effects of stress on the glucocorticoid receptor 1-7 promoter in terms of H3 histone acetylation. I point out, that other than this, these papers do not show much overlap. I have a general sense that the authors have responded to my critique without answering the specific questions directly. In some places, they indicate that these issues were addressed in the introduction. Needless to say, I was not able to find the requested information.
We described our previous study on histone acetylation of GR promoter I7 (Neurosci Lett, 2017; reference [7]) in previously revised manuscript. Thus, the reason we investigated histone modification of the GR promoter I7 was described following this sentence (line 77 to 80 of page 2).
Point 3. I asked for the rationale for looking at H3K9/14ac and H3K4me3. They indicate that most people have looked at DNA methylation while histone modifications are also important. I agree, but was mainly wondering why those particular histone modifications were chosen.
These markers (H3K9/14ac, H3K4me3, H3K27me3) were based on studies on epigenetic mechanisms of depression and antidepressant action. These studies mainly focused on histone modification of BDNF gene, and these markers clearly exhibited that they are markers for regulation of gene expression at BDNF promoters. Thus, we cited representative papers in 4.5 Chromatin immunoprecipitation assay (ChIP assay) of 4. Materials and Methods section (line 331 to 332 of page 9).
Point 4. In response to my concern regarding their ChIP antibodies, the authors have rewritten the methods section and included additional detail. They include the manufacturer of the antibodies used. My question was – did they do any testing of these antibodies to detgermine their specificity or did they assume that the company sent them verified antibodies. They also mention that the magnetic beads are ChIP-grade. However, it’s generally of concern that the antibodies are ChIP grade, not the mag beads.
Antibodies used in ChIP assay were all the ChIP-grade. We tested previously specificity of antibodies.
FIGURE. Chromatin immunoprecipitation were performed using digested chromatin from mouse hippocampal tissue and ChIP-validated antibodies. Purified DNA was analyzed by qRT-PCR, using the SimpleChIP® Mouse RPL30 intron 2 promoter primers #7015 (Control primer set; upper bands, 159 bp) and GR exon I7 promoter primers (lower bands, 216 bp). PCR products were observed for each primer set in the input sample (lane 2) and various ChIP samples (Lane 3-5), but not in the Normal Rabbit IgG ChIP sample (Lane 6).
Lane 1; Size marker (100 bp ladder; EBM-1001, Elpisbio)
Lane 2; input sample
Lane 2; AcH3 (K9/K14; 06-599, Millipore)
Lane 3; H3K4me3 (ab8580, Abcam)
Lane 4; H3K27me3 (ab6002, Abcam)
Lane 5; Normal Rabbit IgG (#2729, Cell signaling)
The authors need to specific which part of the glucocorticoid receptor mRNA they are measuring. Is it the total NR3C1 mRNA or only the NR3C1-7 exon mRNA?
We measured the total GR mRNA but not GR exonI7 mRNA. ‘GR expression’ described in manuscript was corrected to ‘total GR expression’.
Point 5. I had previously requested why they chose to examine HDAC5. The authors did not respond appropriately. Did they only measure mRNA for HDAC5 or others as well. I am not sure that this measurement has any meaning in the absence of protein measurements for HDAC5. If, for example, if their housekeeping gene used for normalization changes in response to stress, the HDAC5 mRNA measurements will be off. If they want to make a connection between the decreased H3ac and HDAC5, it's the protein that would be responsible, note mRNA.
We measured only HDAC5 mRNA among HDAC classes. Although HDAC5 is associated with depression, as reviewer say, HDAC5 mRNA measurements cannot be linked to histone acetylation. Thus, the sentences that connected between HDAC5 mRNA and histone acetylation (line 141 of page 4, line 255 to 256 of page 8) were deleted.
Point 6. Movement of the Figure to Fig. 1 helps with my concerns. This is now clear.
Points 7 and 8 are adequately addressed.
Minor Concerns:
1.In the legend to Fig. 3, changes is mis-spelled.
2.In the legend to Fig. 4, acetylatuion is incorrectly spelled and there is a disconnect- ‘at the of GR.’
3. In Table 1: it should be PCR for histone modification, not RT-PCR.
As suggested, 1~3 were corrected.